# Synergistic effect of CNTF and GDNF on directed neurite growth in chick embryo dorsal root ganglia

Vladimir Mashanov[1]*, Abdelrahman Alwan[1], Michael W. Kim[1], Dehui Lai[1,2], Aurelia Poerio[1,3], Young Min Ju[1], Ji Hyun Kim[1], James J. Yoo[1]

**1** Wake Forest Institute for Regenerative Medicine, Winston Salem, NC, United States of America,
**2** Department of Urology, Fifth Affiliated Hospital of Guangzhou Medical University, Guangzhou, China,
**3** Institut Jean Lamour, Université de Lorraine, Nancy, France

* vmashano@wakehealth.edu

**Data Availability Statement:** All relevant data are within the manuscript and its Supporting Information files.

## Abstract

It is often critical to improve the limited regenerative capacity of the peripheral nerves and direct neural growth towards specific targets, such as surgically implanted bioengineered constructs. One approach to accomplish this goal is to use extrinsic neurotrophic factors. The candidate factors first need to be identified and characterized in *in vitro* tests for their ability to direct the neurite growth. Here, we present a simple guidance assay that allows to assess the chemotactic effect of signaling molecules on the growth of neuronal processes from dorsal root ganglia (DRG) using only standard tissue culture materials. We used this technique to quantitatively determine the combined and individual effects of the ciliary neurotrophic factor (CNTF) and glial cell line-derived neurotrophic factor (GDNF) on neurite outgrowth. We demonstrated that these two neurotrophic factors, when applied in a 1:1 combination, but not individually, induced directed growth of neuronal processes towards the source of the gradient. This chemotactic effect persists without significant changes over a wide (10-fold) concentration range. Moreover, we demonstrated that other, more general growth parameters that do not evaluate growth in a specific direction (such as, neurite length and trajectory) were differentially affected by the concentration of the CNTF/GNDF mixture. Furthermore, GDNF, when applied individually, did not have any chemotactic effect, but caused significant neurite elongation and an increase in the number of neurites per ganglion.

## Introduction

There is a clear need to design efficient ways to improve the limited intrinsic regenerative capacities of the peripheral nervous system to repair large injuries and facilitate adequate innervation of surgically implanted bioengineered tissue constructs. One of the major focuses in the field is to enhance the regrowth of peripheral nerves through the application of extrinsic neurotrophic factors [1]. These factors are expressed endogenously by Schwann glial cells and

**Funding:** Research reported in this publication was supported by an MTEC award (project No. 2017-614-002) The funders had no role in study design, data collection and analysis, decision to publish, or preparation of the manuscript.

**Competing interests:** The authors have declared that no competing interests exist.

other cell types in response to the neural injury, but this expression is often insufficient to sustain full regeneration and reinnervation [2]. In regenerative medicine, it is often desirable to not only enhance the neural regeneration in general, but to specifically direct the growing neurites towards a desired target [3, 4].

The first step in identifying the potential signaling molecules with the capacity to direct neurite growth is to screen them in chemotaxis assays. In spite of the progress in the field, many of the available guidance assays have been showing limitations regarding controlling the precise concentration of signaling molecules in the gradient field and/or are difficult to implement and reproduce as they involve highly specialized equipment or custom devices manufactured in individual labs [3–5]. In this study, we developed a simple guidance assay that allows to assess the effect of neurotrophic factors on neurite outgrowth from dorsal root ganglia (DRGs) using only standard tissue culture materials. We used this technique to quantitatively determine the combined and individual effects of CNTF (ciliary neurotrophic factor) and GDNF (glial cell line-derived neurotrophic factor) on both directed and general neurite growth in chick embryonic DRGs. These two neurotrophic factors were chosen based on our pilot screening experiments (S1 File). In addition, although both factors are known to positively affect the survival and axonal regeneration of both sensory and motor neurons [6–9], their ability regulate the direction of the neurite growth, either individually or in combination, has not been characterized (see below).

CNTF belongs to the interleukin-6 (also referred to as gp130) cytokine family and is produced *in vivo* by astrocytes and Schwann cells [10, 11]. In the nervous system, it functions to support the survival and differentiation of neurons and glia, as well as neurite outgrowth. At the intracellular level, it activates the PI3K/Akt and JAK2/STAT3 pathways [11–14]. Unlike many other neurotrophic factors, CNTF is abundantly expressed in the uninjured peripheral nerves, but is suppressed during neural regeneration. However, a burst of CNTF release from damaged Schwann cells immediately after injury is thought to be a crucial signal for the synthesis of other trophic factors that support regeneration [15]. CNTF has also been shown to have a chemotactic effect on migrating m aicrophages, but its ability to direct neurite growth has not been directly studied yet [16].

GDNF is a member of the transforming growth factor-$\beta$ superfamily that is secreted *in vivo* by glial cells(Schwann cells, astrocytes, and microglia), but also by neurons and denervated skeletal muscle cells [17, 18]. It binds to a multisubunit GFR$\alpha$/RET membrane receptor. The binding of the ligand to the receptor activates the downstream PI3K/Akt and Ras/MAP kinase pathways to eventually regulate a multitude of cellular events, including neurite outgrowth, branching, synapse formation, and neuronal survival [2, 19–21]. The role of GNDF as a chemoattractant is less studied. Even though it has been shown to direct lateral motor column motor axons to their targets in the dorsal limb, its ability to induce directed neurite growth in other neuronal types remains to be demonstrated [19].

Chick and mammalian DRGs have been extensively used as a convenient model to study the effect of different extrinsic treatments on neurite outgrowth, as they can be easily manipulated and easily subjected to quantification, provide consistent and reproducible results, retain the relevant tissue structure allowing proper interactions between neurons and glia and thus generally show better cell viability *in vitro* than pure cultures of dissociated neurons [1, 3, 22, 23].

Here, we show that CNTF and GDNF induce significant directed growth of DRG neurites towards the source of the gradient. This effect is synergistic in nature, as neither of the two factors had chemotactic effect when applied individually. The ability to cause directed growth persists over a wide (10-fold) concentration range of the CNTF/GDNF mix, suggesting that this combination of the neurotrophic factors might be a promising candidate for future *in vivo*

experiments, where it is challenging to control the exact concentration of the exogenous molecules at all times.

## Results

### Guidance plate assay as a simple and efficient technique to monitor directed neurite growth in DRG explants in culture

Our guidance plate assay (described in detail in the "Materials and methods" section below) (Fig 1) proved to be a simple, reproducible, and efficient way to assess the effect of two-dimensional chemotaxis gradients on directed neurite growth. As described below, it provides robust quantitative results. Besides the general tissue culture equipment and reagents, it does not require any additional specialized and/or costly consumables or devices. To set up the assay, we used standard 35 mm cell culture dishes, whose internal volume was separated into two equal halves by inserting a sterile plastic partition (Fig 1A and 1B). The two halves of the plate can then be filled with a hydrogel of the same or different composition, as determined by the experimental design (Fig 1A and 1C). For example, to assess the efficiency of a chemotaxis signal, one half is filled with a gel containing the signaling molecule(s), whereas the other half contains a "neutral" gel lacking the chemoattractant. Negative control plates are prepared by filling both halves of the dish with the "neutral" gel. After the hydrogels on both sides solidify, the partition is taken out, embryonic DRGs are put into the resulting grove and covered with a layer of the "neutral" gel to hold them in place, prevent their desiccation, and allow diffusion of the morphogen(s) across the gap (Fig 1A, 1C, 1D and 1E'). This approach sets up the gradient across the diametral line of the plate, along which the DRGs are placed. After the *ex vivo in vitro* culture, the explants can be conveniently fixed, immunostained, and imaged *in situ* without the need to remove them from the dish.

### CNTF and GDNF induce directed neurite growth when applied in combination, but not individually

We implemented the guidance plate assay, as described above, to assess the ability of CNTF and GDNF to induce directed neurite growth either individually or in combination. For the purpose of this experiment, we added the neurotrophic factors to the collagen hydrogel on one side of the plate at a concentration of 10 ng/mL each and cultured the DRG explants for 48 hours. The concentration and culture duration parameters were chosen based on our pilot studies and literature data [21]. The cultured explants were processed for immunocytochemistry with an anti-neurofilament antibody and imaged using a confocal microscope (Fig 2). To assess whether or not the neurite growth was directed (i.e., whether or not the neural processes grew preferentially towards the source of the neurotrophic factors), three different metrics were calculated, including: (a) the forward migration index, (b) center of mass displacement, and (c) Rayleigh test p-value. To calculate these metrics, a two-dimensional orthogonal coordinate system was superimposed on the micrographs, with the y-axis oriented along the gradient of the neurotrophic factors, towards the source of the morphogens, and the x-axis perpendicular to the gradient.

Forward migration index (FMI) [24] is a measure of the efficiency of growth in a specific direction, which is calculated as a ratio between the most direct growth path to the total path length (Eq 1). In other words, this index indicates how much of the total growth is used to grow along towards the source of the gradient. In the context of our guidance assay, the direct path is the Euclidean (i.e., straight-line) distance between the proximal and distal ends of a neurite, and the total path length is the actual length of a neurite. FMI is calculated separately

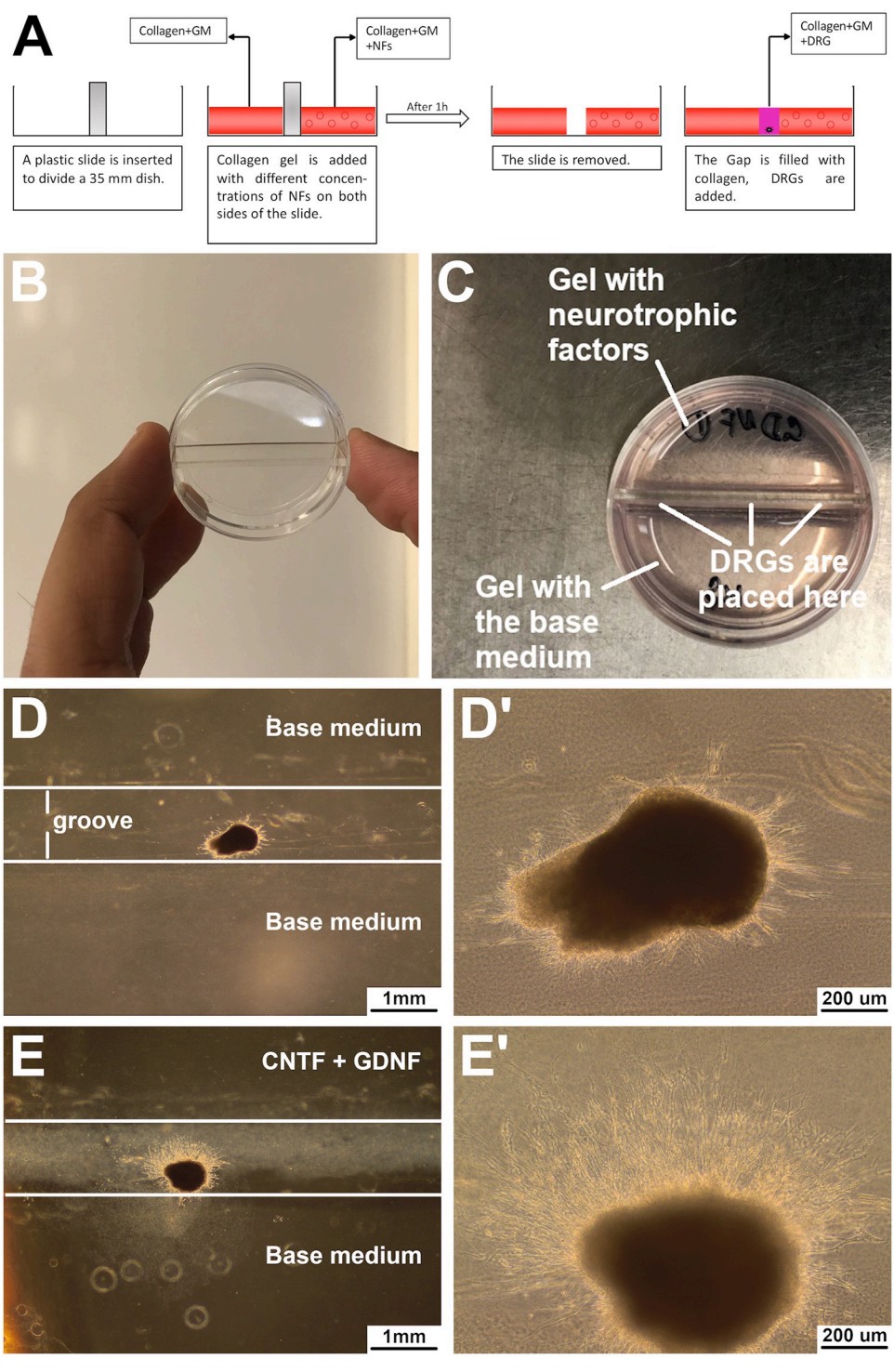

**Fig 1. Guidance plate assay design.** A: Diagram explaining the design. A partition is inserted into a 35 mm dish. One half of the dish is filled with a "neutral" collagen hydrogel containing the growth medium (GM) only. The hydrogel in the other half also contains neurotrophic factors (NFs). After the gels are set, the partition is removed. Dorsal root ganglia (DRGs) are placed in the groove between the two gels and are then covered with the collagen hydrogel. B: Dish with an inserted partition. C: Dish with two different hydrogels set on either side of the partition. D—E': DRGs cultured for 48 hours in a control dish (D and D') that was filled with a hydrogel containing only the base medium on both sides and in a gradient dish with the neurotrophic factors CNTF and GDNF (E and E') added to one of the gels at 10 ng/mL. D' and E': Higher magnification views of the DRGs shown in D and E, respectively.

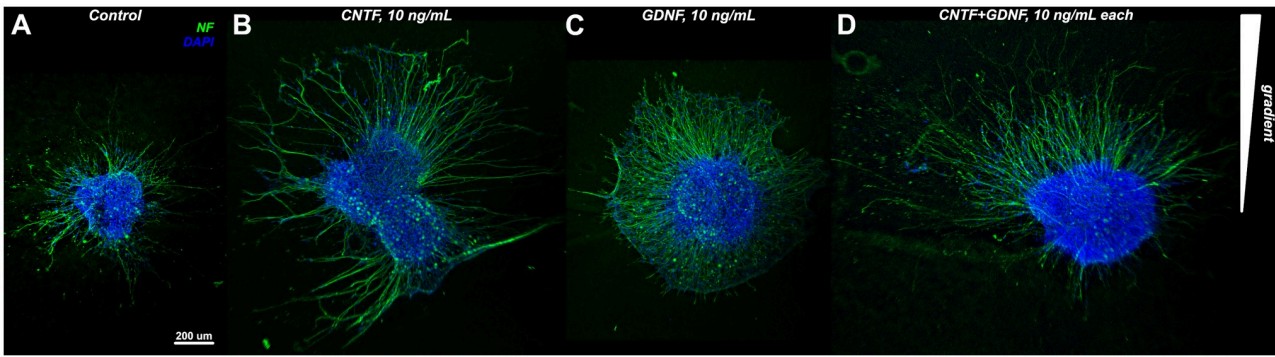

**Fig 2. Representative micrographs of DRGs grown for 48 hours either in control dishes or gradient assay dishes supplied with CNTF and/or GDNF at 10 ng/mL each.** DRGs were stained with antibodies recognizing neurofilaments (NF, green). Nuclei were stain with DAPI (blue). All images represent maximum intensity Z-projections of confocal stacks and are oriented with the source of the neurotrophic gradient at the top. One representative ganglion is shown for each treatment. A: DRG in a control plate cultured in the absence of neurotrophic factors. B—D: DGRs subjected to treatments with CNTF and GDNF either individually or in combination as a source of the chemotaxis gradient: B—CNTF, 10 ng/mL; C—GDNF, 10 ng/mL; D—CNTF and GNDF, 10 ng/mL each.

for the y-axis (YFMI, reflecting growth in the direction of the gradient) and x-axis (XFMI, reflecting growth perpendicular to the gradient) coordinates. In the case of the directed growth of DRG neurites in response to the gradient of neurotrophic factors, the YFMI is expected to be positive and statistically different from the YFMI of the DRGs of the control cohort cultured in the absence of the chemotaxis gradient. On the other hand, XFMI for all cohorts is expected to be close to zero with no significant variation among the cohorts. After 48 days in culture, XFMI did not show any significant variability (One-way ANOVA $F(3, 32) = 0.811$, $p = 0.497$). In contrast, variation in YFMI was highly significant (One-way ANOVA $F(3, 32) = 7.109$, $p = 8.6 \times 10^{-4}$). The use of the combination of CNTF and GDNF as a source of chemotaxis gradient resulted a highly significant increase in YFMI in comparison with the control group ($p = 3.9 \times 10^{-4}$). On the other hand, no significant changes in YFMI were observed in response to treatment with either CNTF or GNDF, when they were applied individually (Fig 3A).

The second metric that we used to determine the ability of CNTF and GDNF to induce directed growth of DRG neurites was center of mass (COM) displacement [5]. This is the measure of the magnitude (in absolute units) at which the neurite ends have extended towards the source of the gradient. The coordinates of the proximal and distal ends of each neurite were transposed so that the proximal ends of all neurites in a given DRG converged at the center of the coordinate plane (x = 0, y = 0). The COM is then calculated as the geometrical average of all the points corresponding to the neurite distal ends (Eq 2). COM displacement from the coordinate plane center thus represents the direction in which the neurites preferentially grow. In the case of the directed growth, COM is expected to shift in the positive direction along the y-axis (YCOM), with no significant displacement along the x-axis (XCOM). The effect of the neurotrophic factor treatment on COM displacement was similar to that on FMI. No significant variation in the COM displacement along the x-axis was observed across the treatment groups (One-way ANOVA $F(3, 32) = 1.357$, $p = 0.247$). The DRGs treated with the combination of CNTF and GDNF showed a significant displacement in the positive direction along the y-axis towards the source of the neurotrophic factors that was significantly higher than that in the control group ($p = 8.2 \times 10^{-5}$) and in the samples treated individually with either CNTF ($p = 1.7 \times 10^{-3}$) or GDNF ($p = 1.4 \times 10^{-2}$). Neither of the latter two cohorts that were treated with the neurotrophic factors individually were statistically different from the control group (Fig 3B).

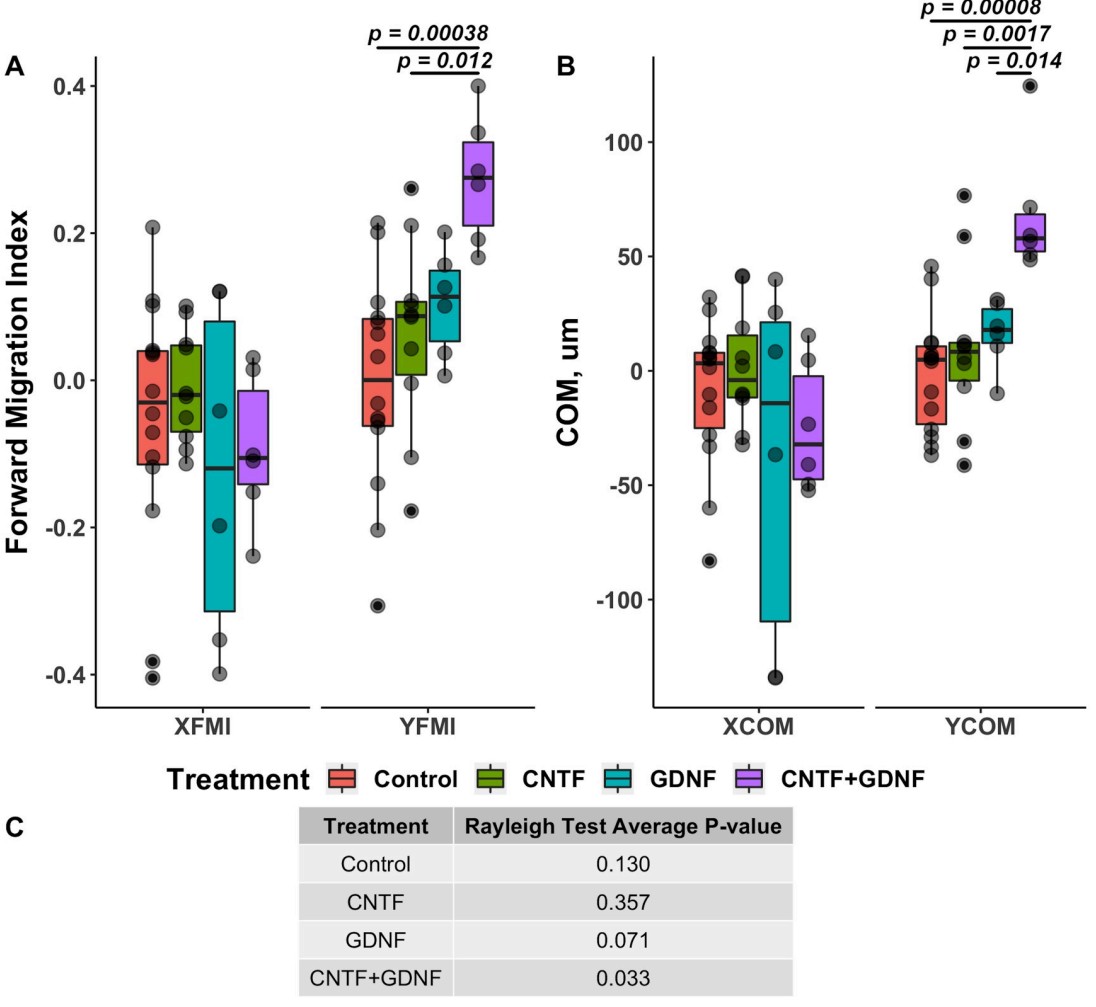

**Fig 3. Quantitative effect of CNTF and GDNF on directed growth of DRG neurites when applied individually or in combination at 10 ng/mL each (48 hours in culture).** A: Forward migration index along the x-axis (XFMI, perpendicular to the gradient of the neurotrophic factors) and y-axis (YFMI, parallel to the gradient). B: Center of mass (COM) displacement in μm along the x-axis (XCOM) and y-axis (YCOM). C: Average *p*-value of Rayleigh test for each treatment group.

The third metric that we used to assess directed neurite growth was Rayleigh test (Eqs 3 and 4), which determines whether or not the coordinates of the distal ends of the growing neurites follow uniform angular distribution [25]. In the case of the directed growth, the null hypothesis of uniformity is rejected with the p-value of the test being below 0.05. After 48 hours in culture, only the cohort, in which the combination of CNTF and GDNF was used as the chemoattractant, had the p-value of Rayleigh test lower than 0.05 ($p = 0.033$). The p-value in the control cohort, as well as in the groups treated separately with CNTF and GDNF was above the threshold level of 0.05 (0.13, 0.36, and 0.071, respectively) (Fig 3C).

Thus, all three metrics that we used to assess the directed growth in response to the neurotrophic factor gradient corroborated each other and showed that CNTF and GDNF have a synergistic effect by inducing the preferential growth of DRG neurites towards the source of the gradient. In contrast, when applied individually, these neurotrophic factors lose the ability to induce directed growth.

In addition to assessing whether or not the DRG neurites exhibited directed growth towards the source of the chemotaxis signal, we also analyzed other growth parameters that do not directly evaluate growth in a specific direction, but may still be affected by the neurotrophic factors. These parameters included: (a) directness, (b) longest neurite length, (c) average neurite length, and (d) number of neurites growing out from a given DRG.

Directness (Eq 5) is a measure of how close trajectories of growing neurites are to the straight line. It is calculated as a ratio of the Euclidean distance between the proximal and distal ends of the neurite to the total length of the neurite. A directness of 1 would indicate that neurites grow along straight-line trajectories. Our data suggests that the use of CNTF and/or GDNF as chemotaxis cues, either in combination or individually at 10 ng/mL each, did not result in any significant changes in the shape of neurite trajectories (One-way ANOVA $F(3, 32) = 1.942$, $p = 0.143$) (Fig 4A).

As to the other metrics, the mixture of CNTF and GDNF (10 ng/mL each) did not cause any changes in the number of neurites growing out from DRGs nor did it affect the maximum length of neurites. However, this combined treatment does significantly ($p = 0.012$) increase the average length of neurites by a factor of ~1.5 as compared to the control group cultured in the absence of neurotrophic factor gradient (Fig 4B–4D). Interestingly, GDNF, but not CNTF, alone at 10 ng/mL caused significant increase in all three parameters (the maximum neurite length, average neurite length, and number of neurites per DRG) when compared to the control group ($p = 0.006$, 0.03, and 0.006, respectively) (Fig 4B–4D).

## The synergistic effect of CNTF and GDNF on directed growth of DRG neurites persists over a wide range of concentrations

After establishing that the combination of CNTF and GDNF at 10 ng/mL each both increases the average length of DRG neurites and induces their directed growth towards the source of the neurotrophic factors after 48 hours in culture, we then asked if this effect will sustain, improve or diminish at higher concentrations of the signaling molecules. To this end, we applied the CNTF/GDNF mix at higher concentrations of 50 ng/ml and 100 ng/mL each as a source of chemical gradient while keeping the duration of the culture at 2 days. We then compared the effect of these treatments on growing DRG neurons with the original treatment when both neurotrophic factors were used at 10 ng/mL and with the control group (Fig 5).

Our data suggests that the combination of CNTF and GDNF sustains directed growth of DRG neurites towards the source of the chemotaxis gradients at all three concentrations. In all three treatments, the preferential extension of the neurites towards the source of the neurotrophic factors was reflected by a significant increase in forward migration index (YFMI) (Fig 6A) and center of mass displacement (YCOM) along the y-axis (Fig 6B) (i.e., parallel to the gradient), as well as average p-values of Raleigh test being below 0.05 (Fig 6C). Interestingly, there is no evidence of enhanced directed growth with increasing concentrations of the neurotrophic factors. Even though YFMI values for all three cohorts treated with CNTF and GNDF were significantly different from the control group, they were not different from each other (Fig 6A). Likewise, the center of mass was found to be significantly displaced in the direction of the gradient source in all three cohorts cultured in the presence of the neurotrophic factors, with the YCOM value in the group treated with the lowest concentration (10 ng/mL) being even higher than the corresponding value in the group treated with the highest concentration (100 ng/mL) (Fig 6B).

As above, we also evaluated the effect of different concentrations of the CNTF and GDNF mixture on the general growth metrics that are not directly related to the directed growth (Fig 7). All four metrics (directness, longest neurite length, average neurite length, and the number

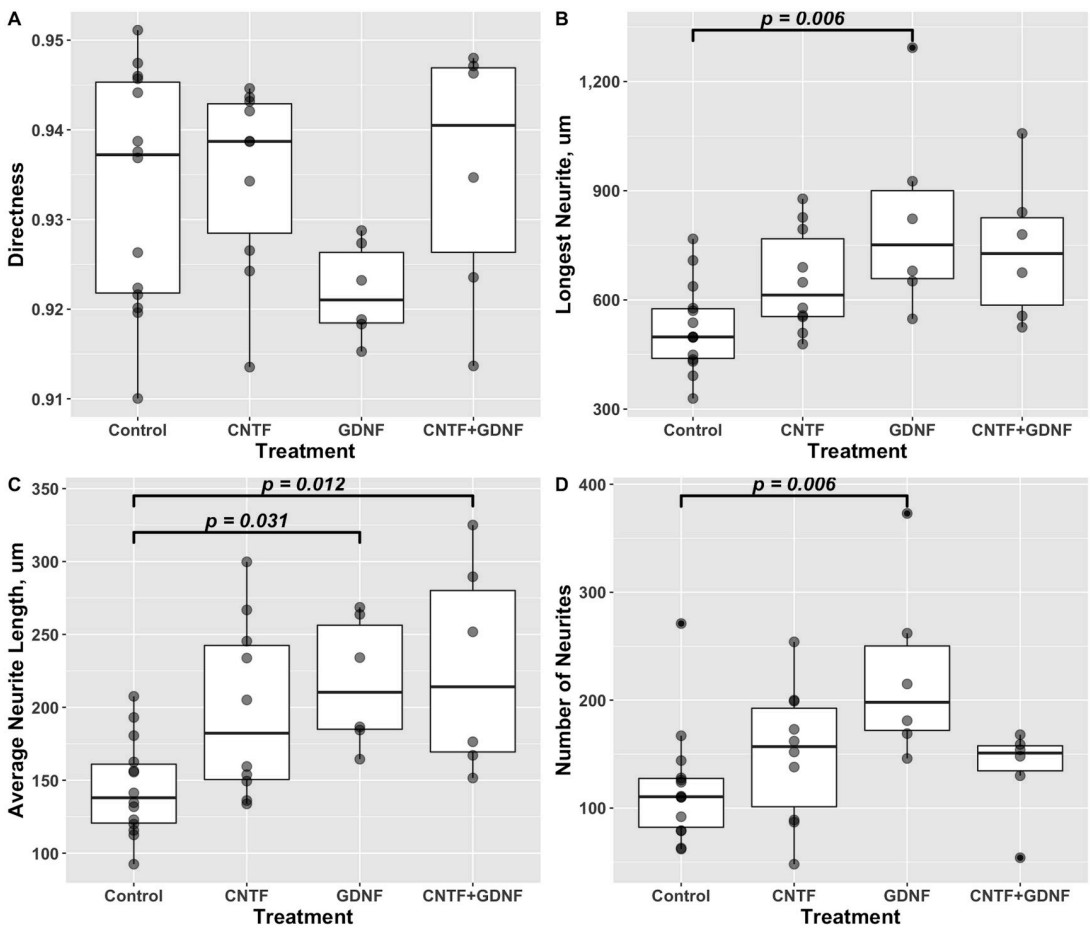

**Fig 4. Quantitative effect of CNTF and GDNF on growth properties of DRG neurites when applied individually or in combination at 10 ng/mL each (48 hours in culture).** A: Directness. No significant variation was detected among the treatment groups (One-way ANOVA $F_{(3, 32)} = 1.942$, $p = 0.143$). B: Length of the longest neurite in DRGs. There was a significant variation among the treatment groups (One-way ANOVA $F_{(3, 32)} = 5.137$, $p = 5.17 \times 10^{-3}$). Tukey post-hoc test revealed that the length of the longest neurite significantly increased in the cohort treated with GDNF compared to the control group. C: Average neurite length. One-way ANOVA detected significant changes among the groups in response to the gradient of neurotrophic factors ($F_{(3, 32)} = 5.318$, $p = 4.35 \times 10^{-3}$). The cohorts exposed to GDNF and the combination of CNTF and GDNF both showed a significant increase in the mean neurite length compared to the control group. D: Number of neurites per DRG. One-way ANOVA showed a significant variation among the treatment cohorts ($F_{(3, 32)} = 4.433$, $p = 1.03 \times 10^{-2}$), with the ganglia cultured in the presence of a GDNF gradient showing a significantly higher number of neuronal processes than the control group.

of neurites per ganglion) differentially reacted to specific concentrations of the neurotrophic factor mix. Even though the combination of these neurotrophic factors had no effect on directness at the lower concentrations (10 ng/mL and 50 ng/mL each), the growing neurites in cultures exposed to the highest concentration (100 ng/mL each) of those signaling molecules deviated from straight-line trajectories more significantly ($p = 0.025$) than in the control group ($0.921 \pm 0.007$ vs $0.945 \pm 0.006$, mean ± SE) (Fig 7A). The longest neurites were significantly ($p = 0.004$) longer only in ganglia exposed to the intermediate concentration (50 ng/mL) of the CNTF/GNDF mix as compared to the control group ($927.3 \pm 107.3$ $\mu$m vs $533.8 \pm 48.3$ $\mu$m, mean ± SE) (Fig 7B). The average neurite length significantly ($p = 0.019$) exceeded the control levels only in the cohort treated with the lowest concentration of the CNTF and GDNF (10 ng/mL) ($226.9 \pm 29.4$ $\mu$m vs $150.7 \pm 9.8$ $\mu$m, mean ± SE) (Fig 7C). The number of neurite growing

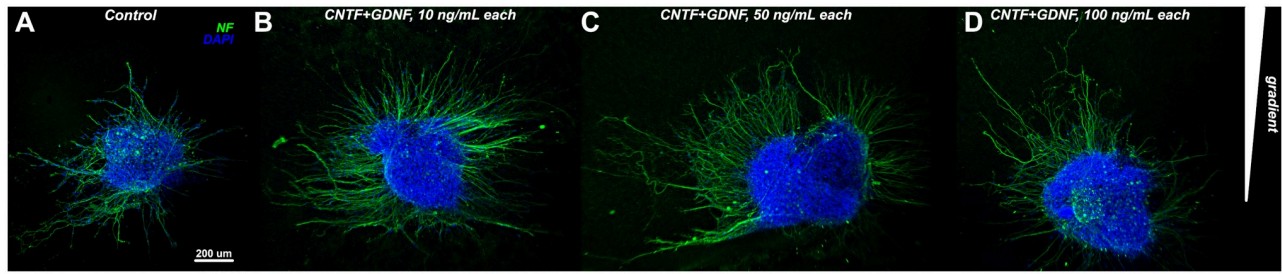

**Fig 5. Representative micrographs of DRGs grown for 48 hours either in control dishes (A) or gradient assay dishes supplied with a mix of CNTF and GDNF at different concentrations (B–D) B: 10 ng/mL; C: 50 ng/mL; D: 100 ng/mL.** DRGs were stained with antibodies recognizing neurofilaments (NF, green). Nuclei were stain with DAPI (blue). All images represent maximum intensity Z-projections of confocal stacks and are oriented with the source of the neurotrophic gradient at the top.

out from a given DRG was significantly ($\sim$2-fold, $p = 0.013$) higher in the cohort treated with the highest concentration of the CNTF/GDNF combination (100 ng/mL) as compared to the control (141.1±21.9 vs 72.9±11.8, mean ± SE) (Fig 7D).

## Discussion

In this study, we developed a simple gradient assay to assess the capacity of chemoattractant signaling molecules to induce directed neurite growth. We used this assay to study the effect of CNTF and GDNF gradients on the growth of neuronal processes in chick embryonic DRGs. Robust directed growth neurite in response to the trophic factors was observed as early as after two days in culture. The advantages of the assay include the ease of the setup that requires only standard tissue culture materials. After a period in culture, the explants can be easily immunostained and imaged *in situ* for the downstream quantitative analysis. The technique can be modified and/or extended as determined by a research question. For example, other signaling molecules and/or combination(s) thereof can be screened; the time in culture can be increased; other explants (e.g., spinal cord), or even cells dispersed in the collagen hydrogel can be studied. Like all *in vitro* asays, our *ex vivo* DRG-based gradient assay has certain inherent limitations, as it does not fully recapitulate all the complexity found at the whole-organism level (e.g., immune response, faster degradation of trophic factors, scarring). Such explant studies are nevertheless valuable as the first approach to identify candidate trophic factors, combinations thereof, as well as their biologically adequate concentrations, which can later be tested in *in vivo* settings.

We found that CNTF and GNDF synergistically induce directed growth towards the source of the chemotaxis gradient, but are not able to do so independently. Synergistic action has been previously reported for various combinations of neurotrophic factors and was shown to increase both neural survival and neurite outgrowth [2, 21, 26]. For example, it was shown that a combined neuroprotective effect of a CNTF/GNDF mixture on axotomized retinal ganglion cells was significantly higher than if both factors functioned in an additive manner [26]. One explanation for this synergistic positive effect is that different neurotrophic factors can activate shared downstream signaling pathways thus enhancing each other's effect and/or operate through complementary signalling axes at the intracellular level. Both these principles might apply to CNTF and GDNF. These two neurotrophic factors are partially redundant as they are both known to activate the PI3K/Akt pathway in target cells downstream of their receptors thus enhancing neuronal survival, neurite growth and sprouting [2, 13, 14, 20]. On the other hand, the two pathways also involve mutually non-redundant intercellular signaling axes, as CNTF also operates though the JAK2/STAT3 pathway [13, 14], while GNDF activates the Ras/

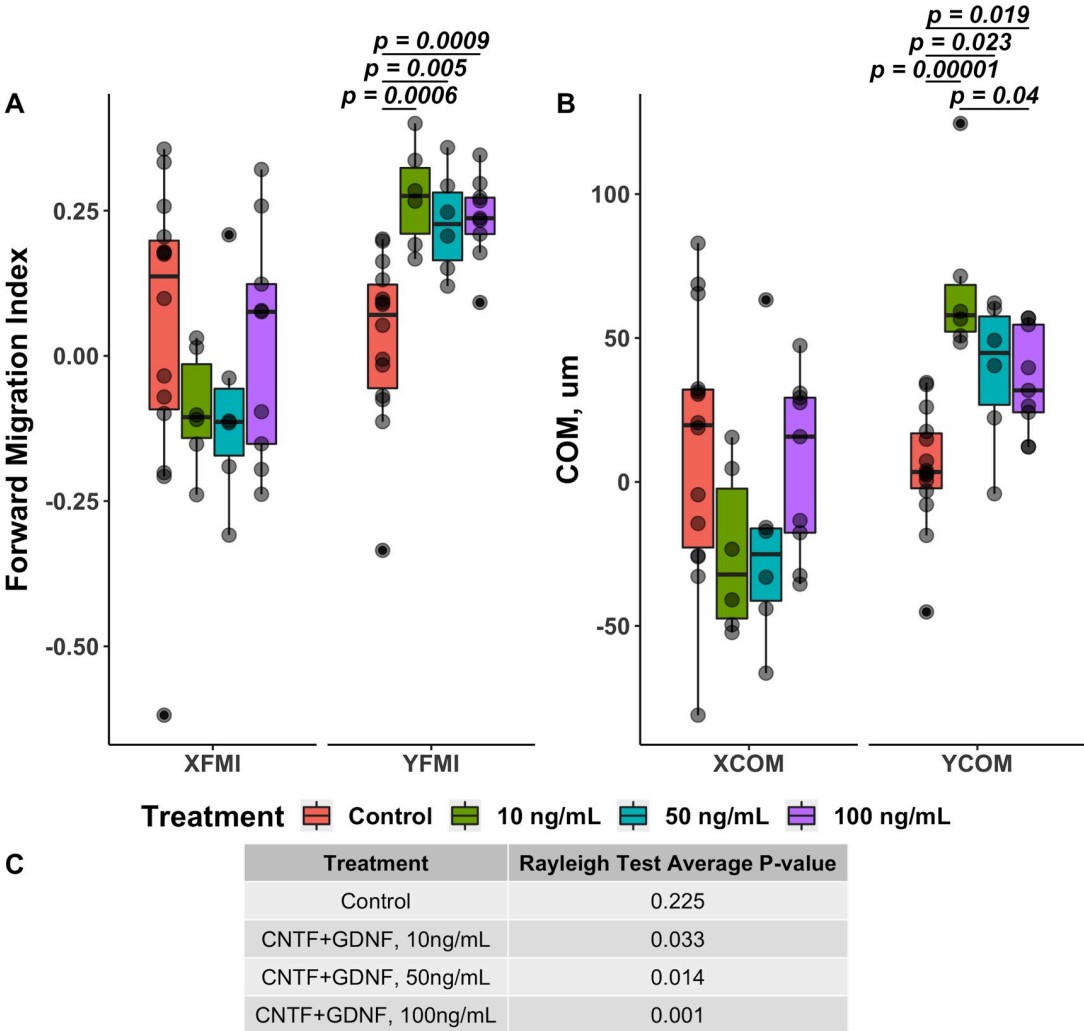

**Fig 6. Quantitative combined effect of different concentrations of the CNTF/GDNF mix (10 ng/mL, 50 ng/mL, and 100 ng/mL each) on directed growth of DRG neurites in the guidance assay after 48 hours in culture.** A: Forward migration index along the x-axis (XFMI, perpendicular to the gradient of the neurotrophic factors) and y-axis (YFMI, parallel to the gradient). No variation in XFMI was observed across the treatment groups (One-way ANOVA $F(3, 31) = 0.893$, $p = 0.456$). YFMI varied significantly (One-way ANOVA $F(3, 31) = 10.42$, $p = 6.71 \times 10^{-5}$) with all three cohorts cultured in the CNTF+GDNF gradient showing higher values than the control group. B: Center of mass (COM) displacement in $\mu$m along the x-axis (XCOM) and y-axis (YCOM). XCOM showed no statistically significant variation among the cohorts (One-way ANOVA $F(3, 31) = 1.691$, $p = 0.19$). YCOM varied significantly among the treatments (One-way ANOVA $F(3, 31) = 12.08$, $p = 2.09 \times 10^{-5}$). All three groups treated with the CNTF+GDNF gradients showed higher YCOM displacement values than the control group. The YCOM values in the group treated with the lowest concentration of CNTF and GNDF (10 ng/mL) were also significantly higher than the values in the group exposed to the highest concentration (100 ng/mL). C: Average p-value of Rayleigh test. Note that only the three cohorts exposed to the gradient of the neurotrophic factors, but not the control group, showed values below 0.05.

MAP kinase pathway [2, 13, 20]. The synergistic effect of the CNTF/GNDF combination on the directed neurite growth does not significantly change within a 10-fold range of the neurotrophic factor mix, suggesting saturation of this response at lower concentrations.

We did find some dose-dependent effects of the CNTF/GDNF mix on the general outgrowth metrics (not directly related to the neurite extension in a specific direction): only the lowest concentration (10 ng/mL) caused the increase in the average neurite length; only the intermediate concentration (50 ng/mL) resulted in the significant elongation of the longest

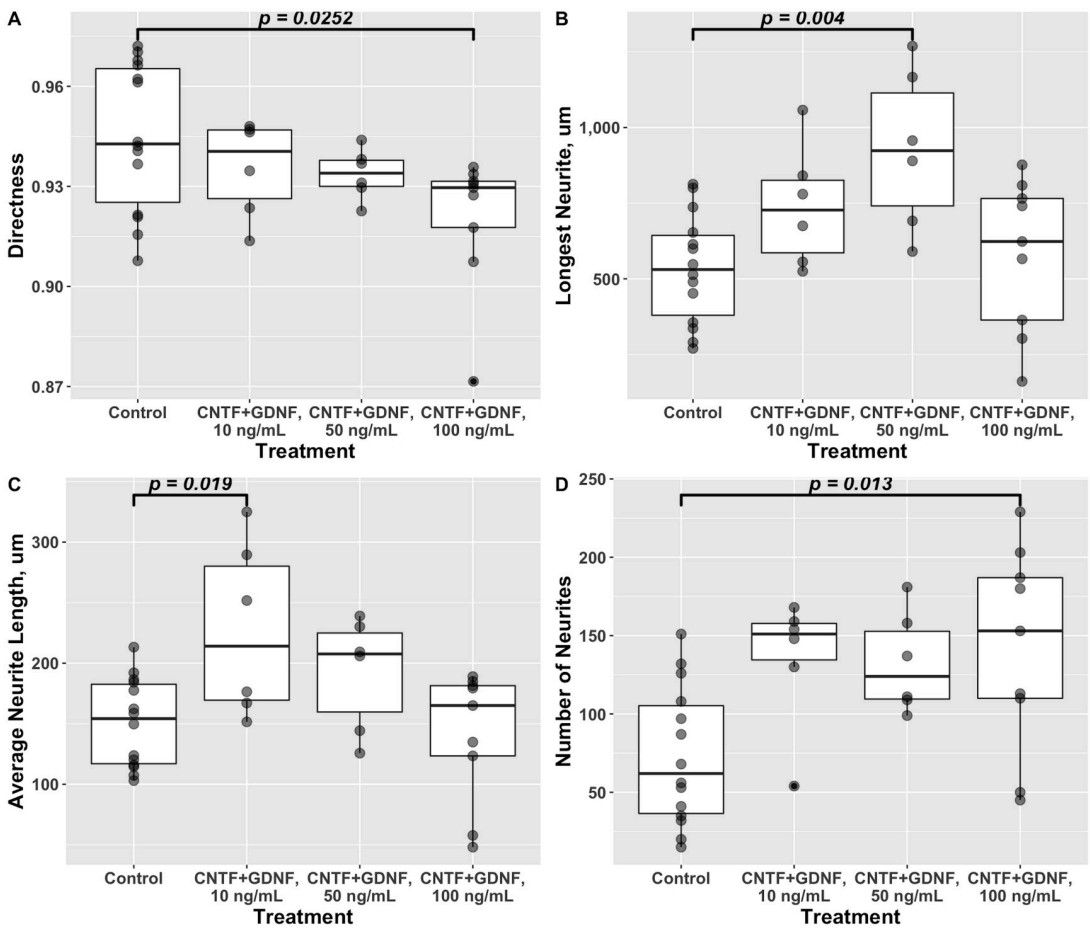

**Fig 7. Quantitative effect of different concentrations of the CNTF/GDNF mix on growth properties of DRG neurites after 48 hours in culture.** A: Directness. Variation among the treatment groups was significant (One-way ANOVA $F_{(3, 31)} = 3.034$, $p = 0.044$) with the group treated with 100 ng/mL of CNTF and GNDF each showing a significantly lower value than the control group. B: Length of the longest neurite in DRGs. There was a significant variation among the treatment groups (One-way ANOVA $F_{(3, 31)} = 5.232$, $p = 4.87 \times 10^{-3}$). The length of the longest neurite significantly increased in the cohort exposed to the intermediate concentration of CNTF/GNDF (50 ng/mL) compared to the control group. C: Average neurite length. One-way ANOVA detected significant changes among the groups ($F_{(3, 31)} = 4.698$, $p = 8.11 \times 10^{-3}$). The cohorts exposed to the lowest concentration of the combination of CNTF and GDNF (10 ng/mL each) showed a significant increase in the mean neurite length compared to the control group. D: Number of neurites per DRG. One-way ANOVA showed a significant variations among the cohorts ($F_{(3, 31)} = 4.865$, $p = 6.9 \times 10^{-3}$), with the ganglia cultured in the presence of the highest concentration of the CNTF/GDNF mix (100 ng/ml) showing a significantly higher number of neuronal processes than the control group.

neurite per ganglion; and only the highest concentration (100 ng/mL) increased the number of neuronal processes growing out from the ganglia and made the neurites to deviate more from straight-line trajectories. Some other neurotrophic factors are also known to exert different, and even opposing, effects depending on their concentrations. For example, exogenous BDNF, due to binding to two different receptors, enhances motor axon regeneration in low doses, but inhibits it at higher concentrations [2]. Both CNTF and GDNF bind to a single receptor complex, therefore the mechanism of the differential dose-dependent action of the mix of these factors is different and further studies are needed to unravel its molecular underpinnings.

Our data also indicate that although GDNF, when applied alone as a source of the chemotaxis gradient, did not cause directed growth of neurites, it still had a positive effect on general growth metrics. More specifically, it induced the increase in both neurite length and their

number without significantly affecting the direction of their growth. These observations corroborate previously published studies that established beneficial effects of this neurotrophic factor on neuronal survival and axonal outgrowth [2, 21, 23, 26].

## Conclusions

We developed a simple gradient plate assay that can be used to assess the capacity of candidate chemoattractants to induce directed growth of neurites from DRG explants. The technique requires only the standard tissue culture materials and can be extended or modified depending on the research question. Using this assay, we studied the chemotactic effects of CNTF and GDNF, that were used as a source of gradient either individually or in combination. Taken together, our data demonstrated the following:

- Exogenous CNTF and GDNF, when applied in a 1:1 combination at 10 ng/mL, but not individually at the same concentration, induce directed growth of neurites towards the source of the gradient.

- The synergistic chemotactic effect of the CNTF/GNDF mixture persists without significant changes over a wide (10-fold) concentration range.

- Although the chemotactic effect of the CNTF/GDNF combination was not affected by the concentration within the tested range (10—100 ng/mL), other—more general growth parameters—showed a differential concentration-dependent response. Only the lowest concentration (10 ng/mL) induced a significant increase in the average neurite length; only the intermediate concentration (50 ng/mL) resulted in an increase of the maximum neurite length per ganglion. Only the treatment with the highest concentration (100 ng/mL) induced the outgrowth of a significantly higher number of neurites from each ganglion and caused the neurites to deviate more from the straight-line trajectories. The underlying mechanism for this dose-dependent effect on the general growth parameters remains to be elucidated.

- GDNF when applied individually did not have any chemotactic effect, but did influence general growth parameters. It caused significant neurite elongation and also an increase in their number per ganglion.

## Materials and methods

### Isolation of dorsal root ganglia

Dorsal root ganglia (DRGs) were dissected from stage 33 (∼E8) chick embryos using sterile fine tweezers according to a protocol adapted from Powell et al, 2014 [27]. The Wake Forest Institutional Animal Care and Use Committee (WF IACUC) provided us a waiver from ethics board review. Since the response of DRGs to identical concentrations or gradients of neurotrophic factors is known to significantly vary depending on their position along the rostro-caudal axis of the body [4], we limited our assays to the ganglia from the lumbar spinal levels. Upon isolation, they were briefly washed in sterile PBS and then immediately collected in the base medium (DMEM/HG with 2% horse serum and 1% penicillin/streptomycin) until plating in collagen gel for the guidance assay.

### Guidance assay

To determine the effect of neurotrophic factors on directional neurite growth, we developed a simple guidance assay (Fig 1). A 35-mm dish is divided into two halves using a sterile plastic

partition (1.5 mm wide × 9 mm high × 35 mm long) that is placed along the diameter of the dish (Fig 1A and 1B). This setup allows casting two hydrogels (with the same or different composition) in the same dish, as described below (Fig 1A and 1C). After the gels are set, dorsal root ganglia are placed in the resulting groove and are covered with the collagen hydrogel (Fig 1A, 1D and 1E').

To prepare collagen hydrogels, rat tail type I collagen (Corning) was mixed on ice with the base medium and 27 $\mu$L of 7.5% sodium bicarbonate per milliliter of added collagen stock solution to a final concentration of 0.2% collagen in the gel [4]. CNTF (R&D Systems, 257-NT-10) and GDNF (Millipore Sigma, GF322) were added to the basal medium either individually or in combination at concentrations of 10 ng/mL, 50 ng/mL or 100 mg/mL. For the guidance assay, 750 $\mu$L of the collagen solution containing the growth factor(s) was added to one half of the plate, and the same volume of collagen prepared in the base medium only was added to the other half of the plate. Before removing the partition and embedding the DRGs, the gels were allowed to set in the incubator at 37˚C for 1 hour. The control plates were loaded with the gel prepared in the base medium and containing no growth factors. After embedding of the DRG explants, the plates were incubated at 37˚C for 2 days with 5% $CO_2$.

## Immunostaining and imaging

The plates were fixed in 10% neutral buffered formalin with 0.1% Triton-X100 overnight at 4˚C. They were then washed 5 × 1 hour in PBS and incubated for 2 hours in the Protein Block solution (Dako, X0909) to minimize non-specific immunostaining. The plates were then incubated overnight in the primary anti-neurofilament heavy polypeptide antibody (Abcam, AB4680) diluted to 1:1,000 in the Antibody Diluent (Dako, D3022), washed 5 times for 1 hour each in PBS, followed by an overnight incubation in the Alexa Fluor 488-conjugated secondary antibody (ThermoFisher, A-11039) diluted at 1:200. The plates were washed as above and mounted in the Vectashield Antifade Mounting Medium with 1.5 $\mu$g/mL DAPI (Vector Laboratories, H-1200) diluted with PBS at a ratio of 1:1. The explants were imaged with a Leica TCS LSI macro confocal microscope.

## Image analysis

Stacks of confocal images were processed with the Fiji/ImageJ software [28, 29]. If necessary, the background (from the collagen hydrogel autofluorescence) was reduced or removed using the "rolling ball" algorithm on individual slices. Maximum intensity Z-projections were then generated for each stack. All micrographs are oriented with the source of the neurotrophic factor gradient at the top. For each neurite, the start and end coordinates, as well as length were obtained using NeuronJ plugin for ImageJ (S2 and S3 Files) by manual tracing of the neural processes in calibrated micrographs [30].

To quantitatively assess the ability of the neurotrophic factors to induce directed neurite growth in the cultured DRGs, we used three different metrics: forward migration index, center of mass displacement, and Rayleigh test.

**Directed growth: Forward migration index.** Forward migration index (FMI) represents the efficiency of the directional growth towards the source of the chemoattractant [24]. Assuming that the gradient of the signaling molecule is established parallel to the Y-axis and the X-axis being perpendicular to the gradient, the FMI for the two axes is (YFMI and XFMI) calculated as follows:

$$YFMI = \frac{1}{n}\sum_{i=1}^{n}\frac{Y_{i,end}-Y_{i,start}}{length_i}; XFMI = \frac{1}{n}\sum_{i=1}^{n}\frac{X_{i,end}-X_{i,start}}{length_i} \qquad (1)$$

where $X_{i,start}$, $Y_{i,start}$ and $X_{i,end}$, $Y_{i,end}$ are the coordinates of the proximal and distal end of each neurite, respectively; and $length_i$ is the total length of the neurite.

**Directed growth: Center of mass.** Center of mass (COM) for each DRG represents the spatial average of the distal end coordinates that have grown from the ganglion. To use this metric, the coordinates for the proximal and distal ends of the neurites were extrapolated in such a way that all proximal ends converged at the center of coordinates (X = 0, Y = 0). The center of mass can then be calculated as:

$$COM = \frac{1}{n}\sum_{i=1}^{n}(X_{i,end}, Y_{i,end}) \tag{2}$$

where $X_{i,end}, Y_{i,end}$ are the transposed coordinates of the distal neurite ends in a DRG.

COM displacement [5] from the center of coordinates then indicates the predominant direction, in which the neurites predominantly have grown from the ganglion.

**Directed growth: Rayleigh test.** Rayleigh test that evaluates the null hypothesis of the uniform circular distribution was performed as described by Moore [25].

Briefly, the vector length ($r_n = \sqrt{(X_{end} - X_{start})^2 + (Y_{end} - Y_{start})^2}$) was calculated for each neurite in a DRG, which is equal to the Euclidean distance between the proximal and distal ends. The vectors corresponding to all neurites produced by a given ganglion are then ranked in an ascending order corresponding to their length. The absolute vector lengths are then transformed by giving the shortest vector the length $r = 1$, the second shortest vector the length $r = 2$, ..., and the longest vector the length $r = N$ (where $N$ is the number of neurites growing from the DRG). If $\theta_n$ is the directional angle of vector $n$, then $\cos\theta_n = \frac{X_{end} - X_{start}}{r_n}$ and $\sin\theta_n = \frac{Y_{end} - Y_{start}}{r_n}$. The test statistics $R^*$ is then calculated in the following steps:

$$X = \sum_{n=1}^{N}n\cos\theta_n; Y = \sum_{n=1}^{N}n\sin\theta_n; R^2 = X^2 + Y^2; R^* = \frac{R}{N^{3/2}} \tag{3}$$

For the ganglia that sprouted 100 or less neurites, the $p$-value of Rayleigh test was determined using the table provided in the original paper [25]. Otherwise, as suggested by the author, it was calculated as follows:

$$p = e^{-3R^{*2}} \tag{4}$$

**General growth metrics.** In addition to the quantitative analysis of the directed growth as above, we also calculated several other metrics, which evaluate neurite sprouting in general, rather than in a predominant direction.

(a) *Directness (D)* is a measure of deviation of neurite trajectories from a straight line. It is calculated as an average ratio between the Euclidean distance between the distal ($X_{i,end}$, $Y_{i,end}$) and proximal ($X_{i,start}$, $Y_{i,start}$) ends of the neurite and the total neurite length ($length_i$) in a given DRG:

$$D = \frac{1}{n}\sum_{i=1}^{n}\frac{\sqrt{(X_{i,end} - X_{i,start})^2 + (Y_{i,end} - Y_{i,start})^2}}{length_i} \tag{5}$$

Three other general growth parameters were calculated for each DRG: (b) *longest neurite length*, (c) *average neurite length*, and (d) *number of neurites*.

Statistical significance was determined with one-way ANOVA followed by Tukey post-hoc test in R [31] and was accepted at the levels of $p < 0.05$ (S4 File). At least 4 DRGs per treatment were analyzed. The plots were generated using the ggplot2 R library [32].

## Supporting information

**S1 File. Description of the pilot screening experiments that identified CNTF and GDNF as the candidate neurotrophic factors to focus on in the present study.**
(PDF)

**S2 File. This file contains the neurite measurements pertaining to the individual and combined effects of CNTF and GDNF (at 10 ng/mL each) on the neurite outgrowth.** The data (in $\mu$m) are shown as exported from the NeuronJ plugin in Fiji/ImageJ.
(XLSX)

**S3 File. This file contains the neurite measurements (in $\mu$m, as exported from the NeuronJ plugin in Fiji/ImageJ) pertaining to the effect of different concentrations of the CNTF/GDNF mixture (10 ng/mL, 50 ng/mL, and 100 ng/mL each) on the neurite outgrowth.**
(XLSX)

**S4 File. This is a representative R code used to process the quantitative data and make plots.** This particular script was used to process the measurements pertaining to the effect of different concentrations of the CNTF/GDNF mixture.
(R)

## Acknowledgments

The authors thank Kenneth Gyabaah for his help with confocal imaging.

## Author Contributions

**Conceptualization:** Vladimir Mashanov, Abdelrahman Alwan, Young Min Ju, Ji Hyun Kim, James J. Yoo.

**Formal analysis:** Vladimir Mashanov, Abdelrahman Alwan, Michael W. Kim, Dehui Lai, Aurelia Poerio.

**Funding acquisition:** Young Min Ju.

**Investigation:** Vladimir Mashanov, Abdelrahman Alwan, Michael W. Kim, Dehui Lai, Aurelia Poerio, Young Min Ju, Ji Hyun Kim, James J. Yoo.

**Methodology:** Vladimir Mashanov, Abdelrahman Alwan, Young Min Ju, Ji Hyun Kim.

**Project administration:** Young Min Ju, Ji Hyun Kim, James J. Yoo.

**Resources:** Young Min Ju, Ji Hyun Kim, James J. Yoo.

**Supervision:** Young Min Ju, Ji Hyun Kim, James J. Yoo.

**Validation:** Vladimir Mashanov.

**Writing – original draft:** Vladimir Mashanov.

**Writing – review & editing:** Vladimir Mashanov, Abdelrahman Alwan, Aurelia Poerio, Young Min Ju, Ji Hyun Kim, James J. Yoo.

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
