## [Decision Letter · Decision Letter 0]

5 Aug 2020

PONE-D-20-20860

Synergistic effect of CNTF and GDNF on directed neurite growth

PLOS ONE

Dear Dr. Mashanov,

Thank you for submitting your manuscript to PLOS ONE. After careful consideration, we feel that it has merit but does not fully meet PLOS ONE’s publication criteria as it currently stands. Therefore, we invite you to submit a revised version of the manuscript that addresses the points raised during the review process.

We look forward to receiving your revised manuscript.

Kind regards,

Kadir Ozkan, Ph.D.

Academic Editor

PLOS ONE

2. Thank you for including the following ethics statement on the submission details page:

'The Wake Forest Institutional Animal Care and Use Committee (WF IACUC) provided

us a waiver from ethics board review'

Please also include this information in the ethics statement in the Methods section of your manuscript.

Reviewers' comments:

Reviewer's Responses to Questions

**Comments to the Author**

1. Is the manuscript technically sound, and do the data support the conclusions?

Reviewer #1: Yes

Reviewer #2: Yes

Reviewer #3 Yes

2. Has the statistical analysis been performed appropriately and rigorously? 

Reviewer #1: Yes

Reviewer #2: Yes

Reviewer #3 Yes

3. Have the authors made all data underlying the findings in their manuscript fully available?

Reviewer #1: Yes

Reviewer #2: Yes

Reviewer #3 Yes

4. Is the manuscript presented in an intelligible fashion and written in standard English?

Reviewer #1: Yes

Reviewer #2: Yes

Reviewer #3 Yes

5. Review Comments to the Author

Reviewer #1: The subject matter under investigation is interesting and important. Moreover, it has not been dealt with in a systematic and novel manner for some time.

The in vitro model is clearly defined and has promise for enabling efficient manipulation for hypothesis testing. Inclusion of growth factors in the hydrogel is also advantage for producing gradient effects while allowing neurite outgrowth to occur. Also, the inclusion of control conditions for assessing trophic factor effects is an advantage. Therefore, the model has broad utility.

Several limitations exist that should be discussed: 1) while chick embryo DRGs are a great model, they do not help address the main problem of regeneration in adults; 2) In reality, neurite outgrowth is impeded by insufficient availability of trophic factors as well as rapid interfering in-growth of fibrous connective tissue. The latter is not a confound in the model but would be in real-life situations; 3) the studies were very short-term, preventing determination of the degree to which target connections was achieved; 4) In (Figs 2 & 5) the effective combined treatments with CTNF?GNDF, it appeared that instead of non-specific radial neuritic growth, "directed" outgrowth was accompanied by ?growth inhibition of other neurites--any explanation?; 5) the model system does not allow for testing of substances that cannot be incorporated and fixed into the hydrogel.

Reviewer #2: The manuscript of Mashalov and co-authors assess the effect of neurotrophins GDNF and CTNF on axonal growth in an ex vivo ganglion model. In the manuscript much attention is paid to the mathematical analysis of the obtained results, since experiments with the primary explant culture is difficult due to its heterogeneity. The manuscript can be published after minor revision.

1. Why did the authors use human growth factors? Even if the results obtained have potential clinical use as an approach in regenerative medicine, the authors need to justify the choice of human recombinant growth factors in the chick ganglion model. Or comparing the species-specificity of these growth factors.

2. In pharmacology, when two substances enhance each other's action in comparison with the total action, this effect is called potentiation, as a special case of synergism. This should be indicated.

3.line 70 – it could be called ex vivo culture (or ex vivo model)

4. It is necessary to substantiate why CNTF and GDNF were chosen. What is the advantage over other combinations of factors? It is known that other growth factors (NGF, VEGFb HGF etc.) also stimulate the regeneration of ganglion axons, but the choice of such a combination in this article is in no way justified.

Reviewer #3 This manuscript focuses primarily on the effects of CNTF (a gp130 cytokine) and GDNF (a growth factor of the TGFbeta family) on directed neurite growth. The authors find no effect on this measure by the two molecules by themselves but a synergistic effect if they are used together. The studies are done on explant cultures of embryonic chick DRGs.

1. Although the study is presented in the context of nerve injury in adult mammals (including humans), as just noted the studies are on embryonic chick ganglia. It would be useful to know if similar effects would be found in (1) adult ganglia and (2) in mammalian ganglia.

2. Furthermore, some discussion should be included if these effects are or are not likely to be limited to sensory neurons.

3. For these reasons, the title should indicate that the studies were done on embryonic sensory ganglia.

4. It should be noted that directed growth out of a ganglion explant is quite different from growth from a proximal nerve after a nerve injury for example after a nerve compression in which the axons grow into the distal nerve segment.

5. Different cell types in embryonic sensory ganglia depend on different growth factors. Since NGF, BDNF, and NT3 are not added to these cultures, one would expect neurons dependent on these growth factors might die during the culture period.

6. The authors measure directed neurite outgrowth by three different methods and interestingly come to the same conclusion with each of them. Nevertheless, given this level of detail, it would be useful for the reader to be told the pros and cons of each method.

7. While the investigators examine the combination of CNTF and GDNF at three concentrations, they only measure the individual factors at the lowest concentration (10 ng), and one wonders if they would get some effects of the factors individually on direct growth at somewhat higher concentrations.

8. There is little discussion of what the source(s) of the factors would be in vivo or whether the authors are proposing that the factors should be administered in vivo and if the latter how would they be administered for example in a patient.

9. It is worth mentioning that CNTF has been shown to decrease in peripheral nerves after axotomy.

10. From the micrographs presented, it is not obvious to this reviewer how the number of neurites emanating from the ganglion were counted (e.g., Fig. 2c) and how the investigators could determine whether the neurofilament stained processes represent single neurites or multiple neurites at the light microscopic level.

11. Although all the measurements were made after 48 h in culture, the authors mention that their cultures could be examined for at least 5 days. Was any measure of cell survival made at this longer time point?

12. Although the authors conclude that the effects they see on directed growth persists over a 10-fold range of factor concentrations, this is not obvious from the micrographs they present. For example, compare Fig. 2d with Fig. 5b and 5c.

13. What do the authors mean when they say that “the assay yielded reliable quantitative data”?

14. It is not immediately obvious why the fact that the factors might “activate shared downstream signaling pathways” would lead to a synergistic effect. Do the authors mean that each factor alone does not reach an effective activation of that pathway?

15. On page 2, second paragraph, microphages should be changed to macrophages.

16. CNTF is a member of a large family of cytokines that includes IL-6, leukemia inhibitory factor, and a number of others. In the Introduction it would be useful to refer to some review article on the effects of cytokines of the IL-6 family on regeneration (e.g., Zigmond, Front Mol Neurosci. 2012 Jan 20;4:62).

6. PLOS authors have the option to publish the peer review history of their article (what does this mean?). If published, this will include your full peer review and any attached files.

Reviewer #1: No

Reviewer #2: No

Reviewer #2: No

---

## [Author Response · Author response to Decision Letter 0]

16 Sep 2020

Dear Dr. Ozkan,

I am attaching the revised version of our manuscript PONE-D-20-20860 “Synergistic effect of CNTF and GDNF on directed neurite growth in chick embryo dorsal root ganglia”.

1. We used the PLOS LaTeX template to prepare the manuscript to ensure that all the style requirements are met. We also made sure that the files are also properly named.

2. We included the ethics statement in the “Materials and methods” section

3. The reference to “the data not shown” was removed. Those results are not critical for the paper and were only mentioned once in the “Discussion” section

Below is the itemized response to the points raised by the three reviewers.

Reviewer# 1

Comment 1 “… while chick embryo DRGs are a great model, they do not help address the main problem of regeneration in adults…”

Comment 2 “ … In reality, neurite outgrowth is impeded by insufficient availability of trophic factors as well as rapid interfering in-growth of fibrous connective tissue. The latter is not a confound in the model but would be in real-life situations …”

 We extended the first paragraph of the Discussion section to address the limitations of the DRG-based assays. We acknowledged that although explant-based assays do not fully recapitulate the complexity of the whole-organism level, they nevertheless are valuable as the first approach to identify candidate trophic factors, combinations thereof and their biologically adequate concentrations, which can later be tested in in vivo settings, including in adult mammals. In addition, we are currently running an in vivo study that has been informed by the results obtained in the present manuscript. Specifically, we are testing the effect of the CNTF/GNDF combination on peripheral nerve regeneration in adult male rats.

Comment 3 “… the studies were very short-term, preventing determination of the degree to which target connections was achieved …”

Our in vivo study mentioned above involves later time points (weeks) and also evaluates functional innervation of target (muscle) cells

Comment 4 “... In Figs 2 & 5 the effective combined treatments with CTNF/GNDF, it appeared that instead of non-specific radial neuritic growth, "directed" outgrowth was accompanied by growth inhibition of other neurites--any explanation? …”

In the present study, we demonstrated preferential growth of DRG neurites towards the source of the CNTF/GDNF mix. The available data does not offer direct mechanistic insight into this phenomenon at a more detailed level. The directional growth can be accompanied by the growth inhibition on the side of the ganglion facing away from the gradient, as suggested by the reviewer, or, alternatively, by re-direction of the proximal segments of the growing neurites before they leave the body of the ganglion. 

Comment 5 “… the model system does not allow for testing of substances that cannot be incorporated and fixed into the hydrogel …”

This comment addresses the general limitation of hydrogel hydrogel-based assay. However, the vast majority of growth factors are hydrophilic and therefore can be incorporated into the hydrogel. 

Reviewer# 2

Comment 1 “… Why did the authors use human growth factors? Even if the results obtained have potential clinical use as an approach in regenerative medicine, the authors need to justify the choice of human recombinant growth factors in the chick ganglion model. Or comparing the species-specificity of these growth factors …”

Recombinant human neurotrophic factors, including CNTF and GDNF, have been extensively used in culture of chick cells and neural tissue explants by other others and proven to work in those systems (see e.g., Xie and Adler, 2000; Fischer et al., 2004 https://doi.org/10.1016/j.mcn.2004.08.007, Volpert et al., 2007 https://doi.org/10.1167/iovs.07-0313, Madduri et al., 2009 https://doi.org/10.1016/j.neures.2009.06.003, Chang et al., 2018 https://doi.org/10.1177/0748730418762152, Wang et al., 2018 https://doi.org/10.1016/j.jneumeth.2018.08.002 among others)

Comment 2 “… In pharmacology, when two substances enhance each other's action in comparison with the total action, this effect is called potentiation, as a special case of synergism. This should be indicated …”

We are not sure the term “potentiation” is directly applicable in the context of the combined effect of CNTF and GNDF on the directed neurite growth. Potentiation refers to enhancement of one agent by another so that their combined effect is greater than the sum of individual effects. The definition thus implies a quantitative, but not quantitative difference, between the individual and combined treatments. In our case, neither CNTF nor GDNF caused directed growth when applied individually. However, the combination of the two trophic factors does result in the directed growth. The difference is therefore qualitative.

Comment 3 “… line 70 – it could be called ex vivo culture (or ex vivo model) …”

That term has been changed as suggested by the reviewer

Comment 4 “… It is necessary to substantiate why CNTF and GDNF were chosen. What is the advantage over other combinations of factors? It is known that other growth factors (NGF, VEGFb, HGF, etc.) also stimulate the regeneration of ganglion axons, but the choice of such a combination in this article is in no way justified …”

In the original manuscript, we did provide some rationale for choosing CNTF and GDNF as trophic factors in our guidance assay (lines 20-22 and 26-28). More specifically, we mentioned that neither of the two factors have been previously sufficiently investigated in terms of their ability to have a chemotactic effect on growing neurites. In addition, we now include a description of our pilot experiments (S1 File) that identified CNTF and GDNF as candidate trophic factors for the present study out of the original set of six factors (CNTF, GDNF, NGF, IGF, FGF, and BDNF) chosen based on a literature analysis.

Reviewer# 3

Comment 1 “… Although the study is presented in the context of nerve injury in adult mammals (including humans), as just noted the studies are on embryonic chick ganglia. It would be useful to know if similar effects would be found in (1) adult ganglia and (2) in mammalian ganglia. …”

 To our knowledge, the present study is the first one to demonstrate the chemotactic effect of the CNTF/GDNF mix on neurite outgrowth. As mentioned above (see our response above to Reviewer#1, Comments 1&2), our chemotactic assay involving chick embryo DGRs was the first step that helped us identify the treatment facilitating directed neurite outgrowth for our subsequent in vivo studies. These studies are currently under way and involve adult mammals (rats). 

Comment 2 “… some discussion should be included if these effects are or are not likely to be limited to sensory neurons …”

 The effects of CNTF and GDNF are not limited to sensory neurons. In fact, it has been widely established in the literature that both factors positively affect the survival and axonal regeneration of both sensory and motor neurons (see, for example, Skaper and Varon, 1986 https://doi.org/10.1016/0165-3806(86)90171-9 , Airaksinen and Saarma, 2002 https://www.nature.com/articles/nrn812, Dubovy et al., 2011 https://dx.doi.org/10.1186%2F1471-2202-12-58, Schaller et al., 2017 https://doi.org/10.1073/pnas.1615372114).

 The relevant information and references were added to the “Introduction” section (lines 22-27)

Comment 3 “… the title should indicate that the studies were done on embryonic sensory ganglia…”

 The title was changed as suggested by the reviewer

Comment 4 “… It should be noted that directed growth out of a ganglion explant is quite different from growth from a proximal nerve after a nerve injury for example after a nerve compression in which the axons grow into the distal nerve segment …”

 We fully agree with the reviewer that the injury paradigm has a big influence on both the progression and outcome of the subsequent regeneration. As noted above (please see our response to Comments #1 &2 by Reviewer 1), after having established the effect of the CNTF/GDNF mix on the directed neurite growth in the present study, we have initiated a series of follow-up experiments to test these factors in in vivo settings using adult rats.

Comment 5 “… Different cell types in embryonic sensory ganglia depend on different growth factors. Since NGF, BDNF, and NT3 are not added to these cultures, one would expect neurons dependent on these growth factors might die during the culture period …”

 Determining the survival of different DRG neuronal types was outside of the immediate scope of the present study, as (a) we focused on the metrics of the directed neurite growth and (b) our neurite guidance assays covered relatively short periods of time (e.g. 2 days). However, the reviewer brings up an important point here, as multiple neurotrophic factors are required to work in cooperation to facilitate the survival of both embryonic and adult neurons. Those interactions can be very complex, depend on the time and cell type and are extensively covered in the literature. 

Comment 6 “…The authors measure directed neurite outgrowth by three different methods and interestingly come to the same conclusion with each of them. Nevertheless, given this level of detail, it would be useful for the reader to be told the pros and cons of each method …”

Forward Migration Index (FMI) is a measure of how much of the growth is actually used to grow in a particular direction.

Center of Mass (COM) Displacement is used to determine the magnitude (in absolute units) at which the neurite ends have extended towards the source of the gradient.

The Rayleigh test is a statistical test of uniformity of the circular distribution of the distal neurite ends. At p < 0.05, the distribution is non-uniform, indicating preferential growth in one direction.

As suggested by the reviewer, the explanation of the metrics was extended in the main text of the manuscript. These three metrics evaluate different parameters of the directed growth and, in the present study, they strongly corroborate each other.

Comment 7 “…While the investigators examine the combination of CNTF and GDNF at three concentrations, they only measure the individual factors at the lowest concentration (10 ng), and one wonders if they would get some effects of the factors individually on direct growth at somewhat higher concentrations …”

Our main goal, in the context of our current and future experiments, has been to identify the neurotrophic factor treatment that would cause directed neurite growth towards the intended target. In this study, we established that the CNTF and GDNF, when applied together at 10 ng/mL, act as a strong chemotactic signal for the growing neuronal processes. This effect is lost when the two neurotrophic factors are applied individually at the same concentration. This is a novel finding. We then proceeded to confirm that this newly established combined effect persists at higher concentrations of the CNTF/GDNF mix to establish the possible range for our ongoing in vivo experiments. We did not think it would be necessary to test individual effects of these factors at higher concentrations as both CNTF and GDNF are well characterized in the literature in terms of their action on different neuronal types at a wide range of dosages and treatment conditions.

Comment 8 “…There is little discussion of what the source(s) of the factors would be in vivo or whether the authors are proposing that the factors should be administered in vivo and if the latter how would they be administered for example in a patient …”

The main in vivo sources of CNTF are glial cells: astrocytes and Schwann cells, in the CNS and PNS, respectively (Richardson, 1994 https://doi.org/10.1016/0163-7258(94)90045-0). GDNF is produced by glial cells (Schwann cells, astrocytes, and microglia), but also by neurons (Duarte Azevedo et al., 2020 https://doi.org/10.3390/jcm9020456) and by skeletal muscle in response to denervation (Lie and Weis, 1998 https://doi.org/10.1016/s0304-3940(98)00434-0). The relevant information has been added to the main text of the manuscript.

As to the in vivo administration of the exogenous CNTF and GDNF, our ongoing experiments (see our response to Comments 1&2 by Reviewer 1 above) involve the development of the implantable delivery system that would release the neurotrophic factors in a controlled and sustained manner. The design and results of those studies will be reported in the upcoming manuscripts.

Comment 9 “…It is worth mentioning that CNTF has been shown to decrease in peripheral nerves after axotomy …”

The relevant information and references have been added to the main text of the manuscript (lines 32 thru 35).

Comment 10 “…From the micrographs presented, it is not obvious to this reviewer how the number of neurites emanating from the ganglion were counted (e.g., Fig. 2c) and how the investigators could determine whether the neurofilament stained processes represent single neurites or multiple neurites at the light microscopic level. …”

The original confocal images are of high enough resolution to distinguish individual neurites. As described in the Materials and Methods section, the images were loaded into the Fiji/ImageJ software, and all individual neurites were manually traced with the NeuronJ plugin at high magnification. The software output includes the number of neurites, their length, as well as the coordinates of the proximal and distal ends.

Comment 11 “…Although all the measurements were made after 48 h in culture, the authors mention that their cultures could be examined for at least 5 days. Was any measure of cell survival made at this longer time point? …”

 CNTF and GNDF are known to have beneficial effects on neuronal survival. However, since the focus of this study was on the effect on the neurite growth, no measurement of cell viability/cell death were performed.

Comment 12 “…Although the authors conclude that the effects they see on directed growth persists over a 10-fold range of factor concentrations, this is not obvious from the micrographs they present. For example, compare Fig. 2d with Fig. 5b and 5c. …”

 We disagree here with the reviewer. The directed growth towards the source of the CNTF/GNDF mix (top of the micrographs) are evident in representative micrographs showing all three concentrations of the neurotrophic factor mix (10 ng/ml – Fig. 2D & Fig. 5B; 50 ng/mL – Fig. 5C; and 100 ng/mL – Fig. 5D). In all those cases, the neurites (stained with an anti-neurofilament antibody [green]) are preferentially directed towards the top of the micrographs. Moreover, our measurements and statistical analysis clearly support the observation that the growth is directed towards the source of the morphogens.

Comment 13 “What do the authors mean when they say that “the assay yielded reliable quantitative data”?”

 We changed the sentence to state that the robust neurite growth was observed as early as after two days in culture (lines 225-227).

Comment 14 “It is not immediately obvious why the fact that the factors might “activate shared downstream signaling pathways” would lead to a synergistic effect. Do the authors mean that each factor alone does not reach an effective activation of that pathway?”

 The exact molecular underpinnings for the synergistic effect of CNTF and GDNF on directed neurite growth are yet to be understood at the mechanistic level, and we did not attempt to probe into this issue in the present study. However, we felt it was necessary to provide a tentative explanation in the “Discussion” section based on the information that is already available in the literature. CNTF and GDNF can act through shared pathways, as well as through independent intracellular signal transduction mechanisms. A separate study is required to pinpoint which of those mechanisms (or may be both) are responsible for the directed growth.

Comment 15 “On page 2, second paragraph, microphages should be changed to macrophages.”

 This spelling error was corrected

Comment 16 “CNTF is a member of a large family of cytokines that includes IL-6, leukemia inhibitory factor, and a number of others. In the Introduction it would be useful to refer to some review article on the effects of cytokines of the IL-6 family on regeneration (e.g., Zigmond, Front Mol Neurosci. 2012 Jan 20;4:62).”

 We included the reference to the above paper, as requested by the reviewer

---

## [Editor Report · Decision Letter 1]

23 Sep 2020

Synergistic effect of CNTF and GDNF on directed neurite growth in chick embryo dorsal root ganglia

PONE-D-20-20860R1

Dear Dr. Mashanov,

We’re pleased to inform you that your manuscript has been judged scientifically suitable for publication and will be formally accepted for publication once it meets all outstanding technical requirements.

Kind regards,

Kadir Ozkan, Ph.D.

Academic Editor

PLOS ONE

---

## [Editor Report · Acceptance letter]

25 Sep 2020

PONE-D-20-20860R1 

Synergistic effect of CNTF and GDNF on directed neurite growth in chick embryo dorsal root ganglia 

Dear Dr. Mashanov:

I'm pleased to inform you that your manuscript has been deemed suitable for publication in PLOS ONE. Congratulations! Your manuscript is now with our production department. 

Kind regards, 

on behalf of

Dr. Kadir Ozkan 

Academic Editor

PLOS ONE